# Socioeconomic Inequalities in COVID-19 in a European Urban Area: Two Waves, Two Patterns

**DOI:** 10.3390/ijerph18031256

**Published:** 2021-01-30

**Authors:** Marc Marí-Dell’Olmo, Mercè Gotsens, M Isabel Pasarín, Maica Rodríguez-Sanz, Lucía Artazcoz, Patricia Garcia de Olalla, Cristina Rius, Carme Borrell

**Affiliations:** 1Agència de Salut Pública de Barcelona, 08023 Barcelona, Spain; mmari@aspb.cat (M.M.-D.); mgotsens@aspb.cat (M.G.); mpasarin@aspb.cat (MI.P.); mrodri@aspb.cat (M.R.-S.); lartazco@aspb.cat (L.A.); polalla@aspb.cat (P.G.d.O.); crius@aspb.cat (C.R.); 2CIBER de Epidemiología y Salud Pública, 28029 Madrid, Spain; 3Institut d’Investigació Biomèdica Sant Pau (IIB Sant Pau), 08041 Barcelona, Spain; 4Department of Experimental and Health Sciences, Universitat Pompeu Fabra, 08003 Barcelona, Spain; 5Department of Pediatrics, Obstetrics, Gynecology, Preventive Medicine and Public Health, Universitat Autònoma de Barcelona, 08193 Barcelona, Spain

**Keywords:** COVID-19, inequalities, geographical area, urban area

## Abstract

*Background*: The objective of this paper is to analyze social inequalities in COVID-19 incidence, stratified by age, sex, geographical area, and income in Barcelona during the first two waves of the pandemic. *Methods*: We collected data on COVID-19 cases confirmed by laboratory tests during the first two waves of the pandemic (1 March to 15 July and 16 July to 30 November, 2020) in Barcelona. For each wave and sex, we calculated smooth cumulative incidence by census tract using a hierarchical Bayesian model. We analyzed income inequalities in the incidence of COVID-19, categorizing the census tracts into quintiles based on the income indicator. *Results*: During the two waves, women showed higher COVID-19 cumulative incidence under 64 years, while the trend was reversed after that threshold. The incidence of the disease was higher in some poor neighborhoods. The risk ratio (RR) increased in the poorest groups compared to the richest ones, mainly in the second wave, with RR being 1.67 (95% Credible Interval-CI-: 1.41–1.96) in the fifth quintile income group for men and 1.71 (95% CI: 1.44–1.99) for women. *Conclusion*: Our results indicate the existence of inequalities in the incidence of COVID-19 in an urban area of Southern Europe.

## 1. Introduction

Previous pandemics have shown patterns of inequality, both at the global scale, with richer countries exhibiting lower rates of disease and mortality, and at the national and municipal level, with worse health indicators in areas with poorer socioeconomic levels [1,2] This knowledge must be considered when planning preventive approaches for the population. Regarding the coronavirus disease 2019 (COVID-19) pandemic, knowledge on risk distribution patterns is useful for designing prevention strategies [3,4]. 

Cities show marked social inequalities, characterized as systematic, unfair, and avoidable differences between social groups. The effect of urban life on health depends to a large extent on how cities are organized and governed. Social factors are known to affect opportunities for good health, which can translate into health inequalities. Health indicators are worse in socially disadvantaged groups, according to social class or gender, for instance [5]. Cities offer a favorable environment for the spread of COVID-19 since contact between people is more frequent than in rural areas, especially in deprived areas, which have worse social determinants of COVID-19, such as working and social conditions. 

Existing knowledge on health inequalities requires that any health problem be analyzed through the lens of inequality. For COVID-19, inequalities have been reported for some countries [4,6,7,8], but there is a need for more evidence to inform the design of policies and programs to deal with the pandemic [9], mainly in urban settings where information on COVID-19 in small areas within the city is usually lacking. Several factors have been found to increase the risk of infection and of developing more serious clinical conditions, for example, living below the poverty line, having no health insurance, depending on public transport, being unable to adopt the conditions of quarantine, being homeless, and living in a small dwelling shared with other people [8,10,11,12]. Job type is also important, since frontline jobs (e.g., supermarket clerk, cleaning staff, or truck driver) prevent workers from adopting the level of confinement that has been more widely recommended [8,13]. This disease has received intense social attention, not only because of the need for infected persons to self-isolate, but also because of the repercussions of confinement as a population-level prevention strategy [14].

Therefore, COVID-19 poses many challenges, especially in cities, and a very important aspect is to know its unequal effects on the population, taking into account axes of inequality such as age, gender, and area of residence, in order to identify vulnerable population groups, information which is also important in setting up policies. In Barcelona, a southern European city with 1.7 million inhabitants, COVID-19 has spread widely, with more than 85,000 cases between February 2020 and 15 January 2021 [15]. The first two waves of the epidemic occurred between January and November 2020. The first wave (March to 15 July) finished after the confinement of the population, but during the summer, a second wave occurred (16 July to November) until new control measures were implemented [16].

As public health services have a very important role both in controlling the epidemic and in preventing new infections, a good understanding of the extent of the disease and its distribution is essential. In addition, the first and the second wave of the epidemic have shown different patterns that need to be studied closely to properly tackle these social determinants.

Therefore, the objective of this paper is to analyze social inequalities in COVID-19 incidence, stratified by age, sex, geographical area, and income as axes of inequality, in Barcelona during the first two waves of the pandemic.

## 2. Methodology

### 2.1. Design, Study Population, and Sources of Information

The design of the study in each wave is ecological. The study population consisted of the non-institutionalized population of Barcelona residents. We collected daily data from the Catalan Department of Health on COVID-19 cases confirmed by laboratory tests during the two pandemic waves: from March 1 to July 15 and from July 16 to November 30 of 2020. During the first wave, diagnostic tests (mainly Polymerase chain reaction—PCR) with good validity [17] were performed, mainly in hospitals, on most severe cases and health workers. However, during the second wave, these tests were extended to all cases and contacts, mainly in primary health care centers, and were also used for massive screening in specific locations (schools, high-risk neighborhoods, etc.).

The addresses of all cases were geocoded using the R libraries CartoCiudad and ggmap [18,19], thus obtaining their geographical coordinates. We used these coordinates to assign to each case its census tract of residence and considered the 1068 census tracts of the 2016 census (the median population of each census tract was 1510). We excluded records with no valid home address to geocode into our study areas (<1%).

To assess socioeconomic status, we used the 2016 personal income index at census tract level (2016 census), obtained from the National Institute of Statistics. The source of information to calculate this index was tax information given by the Spanish population (https://www.ine.es/experimental/atlas/experimental_atlas.htm). Finally, data on the population of Barcelona were obtained from the 2019 municipal census and used to calculate the cumulative incidence.

### 2.2. Indicators and Data Analysis

First, we calculated the cumulative incidence of COVID-19 per 100,000 inhabitants by age and income group for men and women for the two waves, which is a measure of risk (the number of new cases divided by Barcelona’s at-risk population stratified by sex, age, and income group).

To analyze inequalities by census tract in each wave for men and women, we calculated smooth cumulative incidences using the hierarchical Bayesian model proposed by Besag, York, and Mollié (BYM) [20]. We previously used BYM models in European projects, where we studied mortality inequalities in cities of Europe, such as in the INEQ-cities (https://www.ucl.ac.uk/ineq-cities/) [21] or the EUROHEALTHY (http://www.euro-healthy.eu) [22] projects. We have chosen this model to solve the problems derived from the direct use of cumulative incidences in small areas. Since at a small-area level, we have to deal with a low number of cases or high heterogeneity in the number of inhabitants between the different areas, characteristics that can produce unstable estimates of the cumulative incidences. This model takes two types of random effects into account, spatial and heterogeneous: the former considers the spatial structure of the data, while the latter deals with non-structural (non-spatial) variability. In our model, we assigned an intrinsic conditional autoregressive prior distribution (ICAR) to the spatial effect (S), which assumes that the expected value of each census tract corresponded to the mean spatial effect of the adjacent census tracts, and had a variance of σ_S_^2^. The heterogeneous effect (H) was represented by a normal distribution with mean 0 and variance σ_H_^2^. We assigned a uniform distribution U (0, ∞) to the standard deviation σ_S_ and σ_H,_ and a normal vague prior distribution to the regression parameters. In addition, with the aim of quantifying the city census tracts with an excess of incidence, for each census tract, we calculated the probability that its cumulative incidence was greater than that of Barcelona city for men and for women.

Then, to study the effect of inequalities in income on the incidence of COVID-19 for each wave and sex, we introduced the income index of the census tract, categorized into quintiles in the BYM model to obtain the risk ratio (RR) of cumulative incidences in each quintile compared to the richest areas as the reference group. We obtained RR estimates based on the mean of their posterior distribution, along with the corresponding 95% credible intervals (95% CI). All distributions were obtained by using the integrated nested Laplace approximations method (INLA), using the R library INLA [23]. We analyzed all data using R Statistical Software (The R Foundation for Statistical Computing, Vienna, Austria) [24].

## 3. Result

From March to November, there were 61,572 confirmed cases of COVID-19. In the next sections, we describe the main characteristics of each wave.

### 3.1. First Wave

There were 12,927 diagnosed cases of COVID-19 (52.1% were women) during this wave. Men had a cumulative incidence of 759 diagnosed cases per 100,000 inhabitants and women 784. A very uneven pattern related to age (Figure 1A) was observed; among men, incidence clearly showed a gradient with age, whereas this was not the case for women. Women showed increasing cumulative incidence, except for the 65–74 year age group, which had a lower cumulative incidence than the 35–64 year and >75 year age groups. The oldest age group (>75 years) displayed the greatest difference between sexes, with men showing 65% greater cumulative incidence (2060 cases per 100,000 inhabitants) than that of women (1248 per 100,000).

Figure 2 illustrates, for each census tract, the distribution of personal income in 2016, the cumulative incidence of COVID-19, and the probability that the cumulative incidence was greater than that for Barcelona city. The incidence of the disease was higher in some deprived neighborhoods, mainly those in the north for men and women, with a distribution similar to that of personal income. When classifying the cumulative incidence by different geographical censuses tracts according to the personal income index for 2016 (Figure 1B), we observed a progressive trend of cumulative incidence when income decreased. However, the poorest areas showed a small decrease in cumulative incidence. The RR followed the same pattern. In men, there was a progressively increasing RR from the second quintile (1.17; 95% CI:1.02–1.33) to the fourth (1.31; 95% CI: 1.13–1.51), but the RR in the fifth (1.17; 95% CI: 0.99–1.36) broke the trend (Figure 1C). These patterns were similar for women.

### 3.2. Second Wave

There were 48,645 diagnosed cases of COVID-19 (53.3% women) during this second wave. Men showed a cumulative incidence of 2928 diagnosed cases per 100,000 inhabitants and women 2865. In this wave, the age group most affected was 15- to 34-year-olds, followed by 35- to 64-year-olds, and differences between sexes were not as marked as for the previous wave (Figure 3A).

Figure 4 maps, for each census tract, the distribution of personal income in 2016, the cumulative incidence of COVID-19, and the probability that its cumulative incidence was greater than that for Barcelona city. The incidence of the disease was higher in some deprived neighborhoods. When classifying the cumulative incidence in different geographical censuses tracts according to the annual personal income index for 2016, we found a higher incidence of COVID-19 in most deprived areas (cumulative incidence for men: 2238.8 per 100,000 inhabitants in the high-income areas and 3869.9 in the low-income areas; for women these figures ranged from 2195.3 to 3907.6) (Figure 3B). The RR showed a different pattern with respect to the first wave with a clear gradient increasing RR when decreasing the income for men and women; RR were 1.67 (95% CI: 1.41–1.96) in the fifth quintile income group for men and 1.71 (95% CI: 1.44–1.99) for women (Figure 3C).

## 4. Discussion

Here, we have shown social inequalities in COVID-19 incidence in a non-institutionalized population in a city of Southern Europe (Barcelona) that has been significantly affected by the COVID-19 pandemic. Two very distinct patterns have emerged: the first wave, based on hospitalized cases, mainly affected the elderly with the incidence being higher among men, and showed a slight pattern of income-related inequalities. By contrast, the second wave, with a population approach, unveiled higher infection incidence in the young population, with similar cumulative incidence by sex and the relevance of income inequalities. Small areas with lower income were more affected, mainly during the second wave of the pandemic.

### 4.1. Differences by Age Group

To understand the dynamics of the two patterns of COVID-19 impact described here, it is important to be aware that the prevention and control strategy implemented in Spain was typical of a country that was not prepared to tackle a pandemic. During the first wave, the scarcity of diagnostic tests (mainly PCR) forced their nearly exclusive use in more severe patients. Therefore, our analysis of social impact patterns showed that the elderly was the most affected. Notably, during this period, excess mortality was also very high (around 3400 deaths) [15]. During the second wave, Spain was more prepared, and diagnostic tests (mainly PCR and antigen test) were available for the whole population. Therefore, in this second wave, the age group most affected was 15–34 years. The excess mortality of this second wave was lower (around 700 deaths) than that of the first wave.

### 4.2. Gender Inequalities

In the first wave, the cumulative incidence of the disease was highest among women aged <64 years and men aged >64 years. However, we found that women accounted for most cases of COVID-19. The oldest age group (>75 years) of men showed the highest cumulative incidence. In the second wave, the pattern by sex was similar and gender differences were smaller than those of the first wave. Note that women, mainly the youngest women, were more vulnerable to COVID-19 because they are overrepresented in care professions and, hence, are more exposed to infection. For instance, more women work in health and social occupations than men. Moreover, women are usually responsible for family and domestic work and become the primary caregivers of people affected by COVID-19 in their household (especially children and the elderly), a situation that has been amplified during confinement [25,26]. On the other hand, among the oldest, men are more prone to chronic conditions and therefore have more risk of being infected.

The Global Health 50/50 project (https://globalhealth5050.org/the-sex-gender-and-covid-19-project/) has reported that men have more severe cases of COVID-19 and more deaths [27] due to several factors. First, biological differences: women generally have a stronger immune system than men, which could explain their lower susceptibility to infection and, in contrast, their higher incidence of autoimmune diseases [28]. Second, some chronic COVID-19–related diseases (such as chronic respiratory diseases) are more common in men. Third, men have a higher prevalence of smoking and excessive alcohol consumption, which is also related to chronic diseases [27]. Fourth, men are less likely to access health services; for instance, COVID-19 testing is more common among women. In Barcelona, the rates of excess mortality during the two waves was higher among men [15].

On the other hand, measures taken to stop the pandemic might lead to other health effects. For instance, home confinement and the closure of schools and other facilities (e.g., day centers for the elderly) in the first wave would have had a greater impact on women [29]. Moreover, there is evidence that in times of economic precariousness, social instability, and confinement at home, there is an increase in intimate partner violence [30].

### 4.3. Socioeconomic Inequalities

Some low-income areas of the city showed a higher cumulative incidence than more advantaged areas, which highlights the impact of income inequalities on COVID-19 infection risk. This pattern was more marked during the second wave.

The geographic and income-related inequalities in COVID-19 that we have observed in Barcelona have also been described elsewhere [8,31,32,33]. They are linked to living and working conditions, and effectively add to underlying inequalities already present.

First, poorer working conditions can favor transmission: workers from deprived social classes may be forced to return to work or may be unable to follow preventive measures, such as confinement, either because they fear being dismissed or because they have a difficult economic situation. The most vulnerable groups are self-employed workers and those belonging to the informal economy. This is also the case for immigrant workers, who often have precarious, low-paid jobs and who sometimes live in small rooms, share the same room at different times, or have precarious health conditions that encourage infection [34,35].

Second, an urban area such as Barcelona has a high population density, which makes physical distancing more difficult than in rural areas. Moreover, disease transmission is more likely in households that are unable to implement distancing recommendations, especially in cases of cohabitation with a COVID-19-positive person. According to data from the Census of Buildings, the average surface area of housing in Barcelona ranges from 45 m^2^ to over 125 m^2^. Thus, two factors have come together in some of the poorest neighborhoods: smaller apartments and longer working hours during the days of confinement, because these jobs allow fewer opportunities to work from home [36]. In addition, the use of public transport during this period in Barcelona has diminished more among people living in affluent areas than among those living in the most disadvantaged ones [37].

Third, people from more disadvantaged social classes have more chronic disorders (hypertension, cardiovascular diseases, diabetes, etc.) and are therefore more vulnerable to COVID-19 [38]. Furthermore, communication and dissemination of messages on disease prevention and containment measures may not reach the whole population uniformly, and may be understood and interpreted differently. This is associated with the social gradient, which also exists in health knowledge between groups according to educational level, social class, and age [39].

Strikingly, in the first wave, some areas of the city with a low socioeconomic level, such as Ciutat Vella in the city center, did not have a high COVID-19 incidence. The population of this district is younger than that of other districts, so the disease may have been less severe and, therefore, cases of COVID-19 might not have been detected because they were not PCR-tested. This may explain the lower RR in the poorest quintile of the population during the first wave. Our hypothesis was confirmed when analyzing the second wave; when massive tests were undertaken, this same district was one of the hardest hit by the pandemic.

It is worth mentioning that many hours of confinement or quarantine in inadequate housing can cause other adverse health effects, as for example, on mental health. Although physical distancing is justified, the long-term health consequences for the general population are difficult to predict. However, we do know that social isolation not only affects mental health but can also affect other chronic diseases as well as different health problems [40]. People living in the most disadvantaged conditions experience the greatest social and economic consequences, with possible implications for their health. Furthermore, the economic consequences of illness, such as job loss and income loss, mostly affect people from the more disadvantaged social classes, consequently having a negative impact on their health [41].

Finally, although health care is not the main determinant of health, the COVID-19 crisis has brought about important changes in the Spanish national health system in order to be able to treat all patients, which has certainly reduced mortality from this disease. However, it is necessary to assess the impact of these changes on access to other types of care, and whether this aggravates the existing inequalities.

### 4.4. Limitations

In our analysis, we excluded people working or living in nursing homes because their addresses are those of the nursing home and so do not necessarily reflect the socioeconomic position of the residents themselves. As mentioned above, the two waves of the epidemic are not strictly comparable, but we considered it relevant to show the two different patterns revealed by data analysis when two different approaches were implemented. Importantly, these two different waves have implied different steps of public health policies at the city level.

In addition, we have analyzed socioeconomic inequalities in COVID-19 at the small area level. Studies of small areas allow us to obtain the cumulative incidence of COVID-19 with a high level of spatial disaggregation, and to reflect spatial patterns that would be hidden in larger areas, such as neighborhoods.

## 5. Conclusions and Recommendations

Our results indicate the existence of social inequalities in the incidence of COVID-19 by age group, gender, geographical area, and income. Due to the complexity of the determinants of health and inequality, the impact of COVID-19 pandemic will require different studies and approaches, but it is essential that they be performed from the analysis of the axes of inequality to draw attention to the different existing patterns and intersections [14].

The results reported here allow us to assess needs in the territory, prioritize specific areas of action, and develop specific interventions from the public health department and other areas (e.g., social or economic). The development of selective policies for specific areas is important in reducing health inequalities, and these policies must complement universal policies [42]. The Municipal Council of Barcelona has already implemented programs targeting these areas, with the objective of facilitating the confinement of the population and disseminating messages to prevent the infection [43].

We would like to raise awareness about the fact that this pandemic could exacerbate social inequalities and, thereby, health inequalities due to the effects of COVID-19, the confinement measures, and the economic and social impacts that are resulting from the pandemic [4]. Therefore, in the near future, inequalities emerging from this pandemic will need to be closely monitored.

## Figures and Tables

**Figure 1 ijerph-18-01256-f001:**
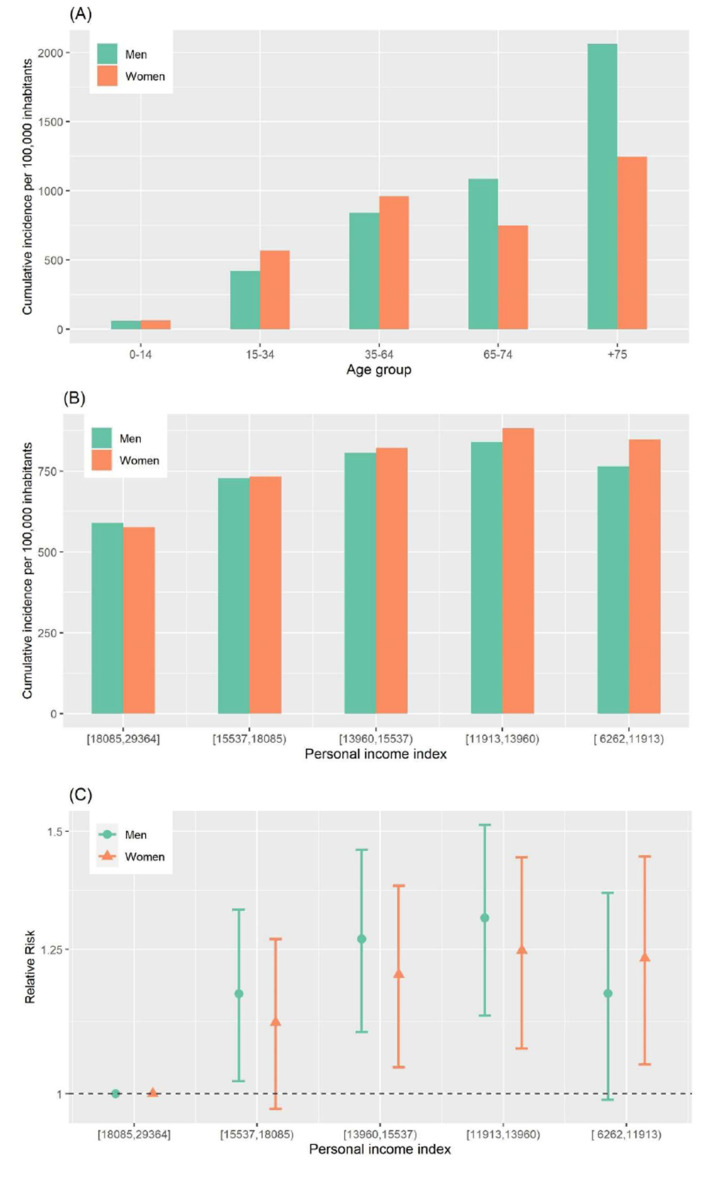
(**A**) Cumulative incidence of COVID-19 per 100,000 inhabitants for each age group and sex; (**B**) Cumulative incidence of COVID-19 per 100,000 inhabitants for each income group (based on personal income index in Euros) and sex; (**C**) Relative risk for each income group and sex. Period from 1 March to 15 July 2020 in Barcelona city.

**Figure 2 ijerph-18-01256-f002:**
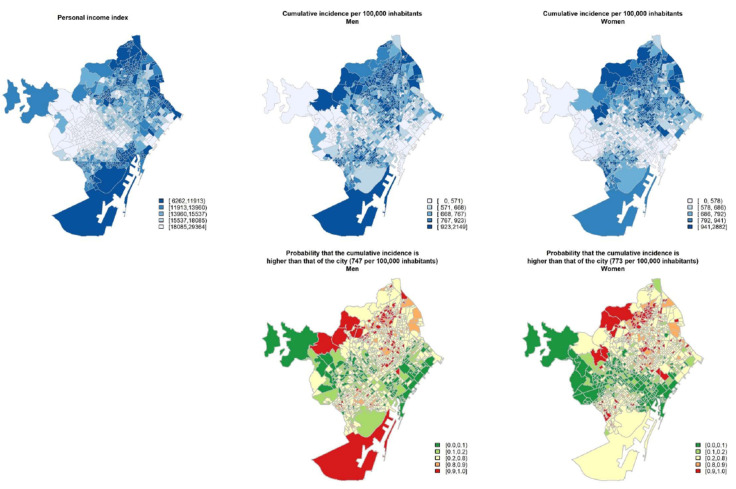
Geographical distribution (at census tract level) of 2016 personal income index, cumulative incidence of COVID-19 per 100,000 inhabitants according to sex and probability of the cumulative incidence being higher than that of the city. Period from 1 March to 15 July 2020 in Barcelona city.

**Figure 3 ijerph-18-01256-f003:**
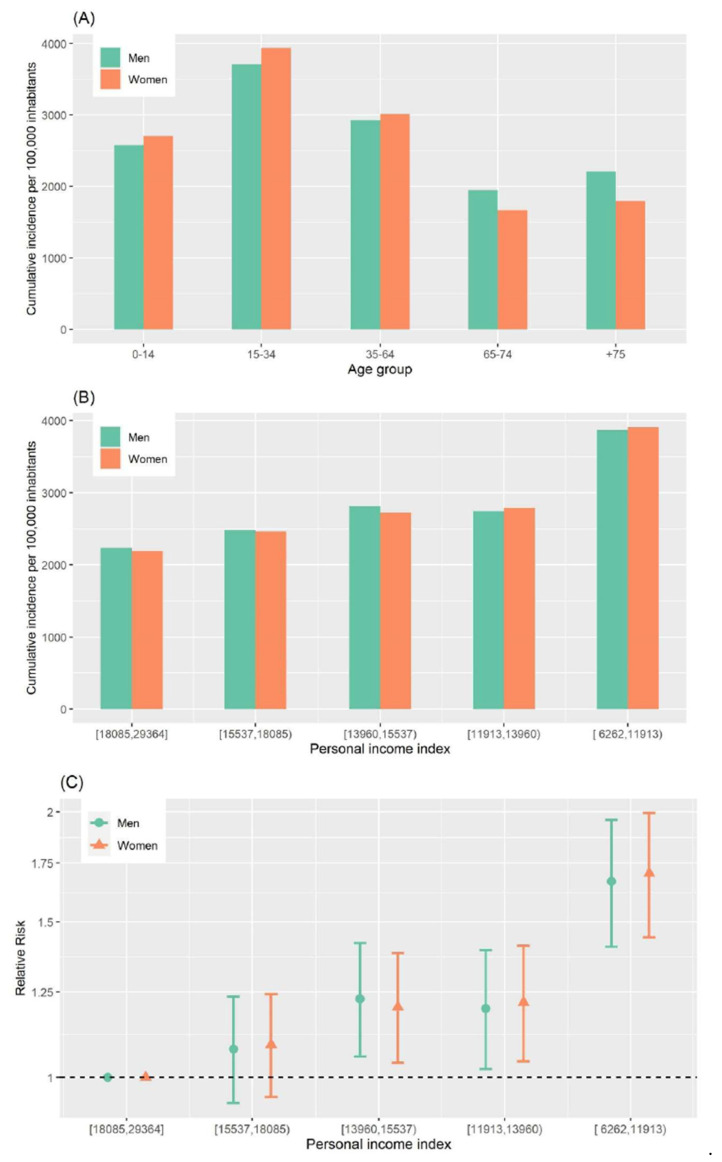
(**A**) Cumulative incidence of COVID-19 per 100,000 inhabitants for each age group and sex; (**B**) Cumulative incidence of COVID-19 per 100,000 inhabitants for each income group (based on personal income index in Euros) and sex; (**C**) Relative risk for each income group and sex. Period from 16 July to 30 November 2020 in Barcelona city.

**Figure 4 ijerph-18-01256-f004:**
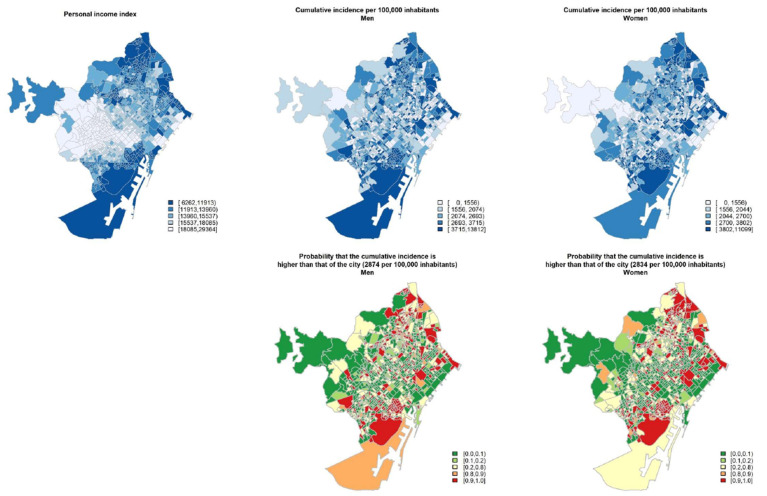
Geographical distribution (at census tract level) of 2016 personal income index (in Euros), cumulative incidence of COVID-19 per 100,000 inhabitants according to sex, and probability of the cumulative incidence being higher than that of the city. Period from 16 July to 30 November 2020 in Barcelona city.

## Data Availability

Data available upon request.

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
