# Peer review of "Socioeconomic Inequalities in COVID-19 in a European Urban Area: Two Waves, Two Patterns"

_ijerph, 2021, doi:10.3390/ijerph18031256_

Round 1

Reviewer 1 Report

This is a observational study discussing about whether there was socioeconomic inequalities in COVID-19 in a European urban area. I think the topic is interesting and contributive to the epidemiology under an empirical approach quite valuable for public social policy professionals.

Major concerns:

  1. Introduction: The author should provide citation to support evidence for “In Barcelona, a southern European city with 1.7 million inhabitants, COVID-19 has spread widely, with more than 70,000 cases since February 2020.”
  2. Introduction: Does any socioeconomic inequalities known for SARS or MERS pandemic?
  3. Method: How is the accuracy of diagnostic tests (mainly PCR)?
  4. Method: It seems poor structured section of Method? The author should clarify this concern.
  5. Method: Why BYM model used in this study?
  6. Results: Which reason caused two waves COVID pandemic? The author should contribute more information to readers.
  7. Results: There still some confounding factor may disturb the results. For example, area location, income, and so on. Any adjusted model for BYM model?
  8. Discussion: Social determinants of health inequalities associated with disease were well known in previous studies. What’s the solution for socioeconomic inequalities in COVID-19 in a European urban area? The author should clarify this concern.

Author Response

ANSWERS TO REVIEWER 1

We enclose our answers to reviewer comments. In the text of the article, changes are shown in yellow colour and new citations in blue colour.

“This is a observational study discussing about whether there was socioeconomic inequalities in COVID-19 in a European urban area. I think the topic is interesting and contributive to the epidemiology under an empirical approach quite valuable for public social policy professionals.”

ANSWER: Thank you very much for your comments, they have been useful to improve the paper.

 Major concerns:

  1. Introduction: The author should provide citation to support evidence for “In Barcelona, a southern European city with 1.7 million inhabitants, COVID-19 has spread widely, with more than 70,000 cases since February 2020.”

ANSWER: We have added a web page.

  1. Introduction: Does any socioeconomic inequalities known for SARS or MERS pandemic?

ANSWER: We have added 2 citations in the first paragraph.

  1. Method: How is the accuracy of diagnostic tests (mainly PCR)?

ANSWER: These tests have good validity. We have added a text and a citation.

  1. Method: It seems poor structured section of Method? The author should clarify this concern.

ANSWER: We have changed the 2 subsections a little bit, now they are: 

2.1. Design, study population and sources of information

2.2. Indicators and data analysis

  1. Method: Why BYM model used in this study?

ANSWER: For this study, a very small area such as the census tract (with 1568 inhabitants on average) has been used as the unit of analysis. This implies that we have obtained Cumulative Incidence of COVID-19 with a high level of spatial disaggregation, so it is possible to represent maps with a high level of "resolution", capable of reflecting spatial patterns that with larger areas, such as neighborhoods, would be hidden. Therefore, they avoid in part the Modifiable Area Unit Problem (MAUP). In addition, this type of ecological design studies are the ones that come closest to the individual level, thus avoiding, to a greater extent, falling into the ecological fallacy.

However, estimates at the small area level can have several problems, on the one hand, a low number of cases in the areas and, on the other hand, a great heterogeneity in terms of the number of inhabitants in the different areas. These characteristics can be an inconvenience in the calculation of the cumulative incidences since areas with low population tend to present very variable estimates. In order to solve the problems derived from the direct use of the Cumulative Incidences, several alternatives have been proposed, such as the Besag, York & Mollié (BYM) model.

Now, we have added a text in the Methods section and in the Limitations section justifying a little bit more the use of BYM models and also these advantages of our analysis.

  1. Results: Which reason caused two waves COVID pandemic? The author should contribute more information to readers.

ANSWER: We included an explanation in the Introduction section referred to the 2 waves and also a citation.

  1. Results: There still some confounding factor may disturb the results. For example, area location, income, and so on. Any adjusted model for BYM model?

ANSWER: We agree with the reviewer’s comment. In relation with the area location, as discussed above (question 5), we have used BYM models to avoid some of the problems derived from using small areas as a unit of analysis. These models take into account the spatial dependence of the data, instead of using the location of the areas, taking into account the information from neighbouring areas to estimate the cumulative incidences of each area. Since, the smooth cumulative incidences of contiguous or more spatially close areas are more similar than the smooth cumulative incidences of more distant areas. As regards other confounding factors, census tracts are a territorial unit that is mainly used as an electoral tool, and it is not easy to find variables at this spatial level. Nevertheless, the analyses have always been carried out separately according to sex and we have obtained a good indicator of socioeconomic deprivation such as income to determine the socioeconomic inequalities in the COVID-19.

  1. Discussion: Social determinants of health inequalities associated with disease were well known in previous studies. What’s the solution for socioeconomic inequalities in COVID-19 in a European urban area? The author should clarify this concern.

ANSWER: We have added a text explaining the advantages of having this information for an urban area with 2 citations in “Conclusions and Recommendations” section.

Reviewer 2 Report

Please see the report in the following file.

Author Response

ANSWERS TO REVIEWER 2

We enclose our answers to reviewer comments. In the text of the article, changes are shown in yellow colour and new citations in blue colour.

“This is an important and urgent topic and the authors executed the study rigorously and provided sufficient information on methods. In the discussion, the authors also provided fair and sound explanations for the observed inequalities. Limitations were well noted as well. My suggestions in the introduction may improve clarify/rigor and my suggestions on method may improve accessibility to wider audience:”

ANSWER: Thank you very much for your comments, they have been useful to improve the paper.

Introduction

- In the 3rd paragraph, please unpack "For COVID-19, this has been shown in some coun-tries2,4–6".

ANSWER: We have tried to rewrite the sentence.

- fourth paragraph: authors noted, "Therefore, COVID-19 poses many challenges, especially in cities, and a very im-portant one is to know its unequal effects in the population.". Here, perhaps tie this back to the point in the 1st paragraph "knowledge on risk distribution patterns is useful for designing prevention strategies". For example, one might say we can identify vulnerable population with repsect to age, gender, and linving area, which in tun can influence policies and etc.

ANSWER: We have included this aspect in the paragraph.

- fifth paragraph: author noted, "In Barcelona, a southern Eu-ropean city with 1.7 million inhabitants, COVID-19 has spread widely, with more than 70,000 cases since February 2020." After this, consider showing a reference, for example, rate at which COVID19 was spread in non-urban area in Spain.

ANSWER: We have added a citation for this information as reviewer 1 asked.

- What is the scientific basis of stratifying the analyses by age, sex, geographical area and income? Please justify selection of these factors, for example, whether they are from prior COVID19 studies in other urban areas or prior pandemics. Method

ANSWER: These factors are important axes of health inequalities, for this reason we have used them in the analysis. We have added this aspect in the objective of the study and also in the previous sentence added in the introduction.

- I am not familiar with the BYM model, and this may be the case for many readers. Please justify/educate the readers on the use of this model over more widely used ones. For example, in multivariate regressions, researchers usually test an interaction with exposure and socioeconomic factors (e.g., gender) controlling for covariates, and then, stratify the analyses. However, such steps are not taken. Is this the nature of BYM? If so, consider explaining this as it may help accessibility of this paper to both health scientists and social scientists.

ANSWER:  As it has been also commented to the reviewer 1 (points 5 and 7), working with small areas as unit of analysis has different advantages: these areas allow to obtain a great spatial disaggregation very useful to reflect the true spatial pattern of the disease and these areas are the closest to the individual level (avoiding to a greater extent the ecological fallacy). Therefore, this type of disaggregation is very useful for disease surveillance and control. However, in order to produce stable estimates of cumulative incidence we must make us of more sophisticated models such as BYM models. These models have the peculiarity that they incorporate two random effects to take into account the variability of the estimates when we work with small areas, but they follow the same logic of analysis as any other statistical model in terms of the inclusion of explanatory, confusing or interacting variables. On the other hand, it is not easy to have variables at the small area level, but even so we have adjusted models with the income variable to study socioeconomic inequalities in the COVID-19. In addition, we have stratified the analyses by sex in order to take into account the inequalities by gender (social aspects related with the biological sex).

 We have added a text in the Methods section and in the Limitations section explaining a little bit more the BYM models and also this advantages of our analysis.

  • At the end of the 2nd paragraph, there is a typo: ",.".

ANSWER: Corrected.

Round 2

Reviewer 1 Report

All concerns had been addressed. 

However, the new references should be inserted and rearranged before publishing. 

The higher-resolution figures required modification to meet IJERPH's quality. 

Congrats!